# Isolation, Characterization, and Application of *Clostridium sporogenes* F39 to Degrade Zearalenone under Anaerobic Conditions

**DOI:** 10.3390/foods11091194

**Published:** 2022-04-20

**Authors:** Congning Zhai, Yangguang Yu, Jun Han, Junqiang Hu, Dan He, Hongyin Zhang, Jianrong Shi, Sherif Ramzy Mohamed, Dawood H. Dawood, Gang Wang, Jianhong Xu

**Affiliations:** 1School of Food and Biological Engineering, Jiangsu University, Zhenjiang 212013, China; zhaicongning2022@163.com (C.Z.); yuyangguang1997@163.com (Y.Y.); zhanghongyin126@126.com (H.Z.); shiji@jaas.ac.cn (J.S.); 2Jiangsu Key Laboratory for Food Quality and Safety-State Key Laboratory Cultivation Base, Ministry of Science and Technology/Key Laboratory for Agro-Product Safety Risk Evaluation (Nanjing), Ministry of Agriculture and Rural Affairs/Key Laboratory for Control Technology and Standard for Agro-Product Safety and Quality, Ministry of Agriculture and Rural Affairs/Collaborative Innovation Center for Modern Grain Circulation and Safety/Institute of Food Safety and Nutrition, Jiangsu Academy of Agricultural Sciences, Nanjing 210014, China; 15150537050@163.com (J.H.); 2021216027@stu.njau.edu.cn (J.H.); danhe58@163.com (D.H.); 3Food Industries and Nutrition Research Institute, Food Toxicology and Contaminants Department, National Research Centre, Tahreer St., Dokki, Giza 12411, Egypt; sheriframzy4@gmail.com; 4Department of Agriculture Chemistry, Faculty of Agriculture, Mansoura University, Mansoura 35516, Egypt; dhosni1978@yahoo.com

**Keywords:** zearalenone, *Clostridium sporogenes*, anaerobic, biodegradation, mycotoxin, feedstuff

## Abstract

Zearalenone (ZEN) is produced by *Fusarium* spp. and is widely found in moldy wheat, corn, and other grains. ZEN has a strong toxicity and causes reproductive and immune disorders and estrogenic syndrome in animals and humans. Biodegradation has been demonstrated as an efficient way to control the hazardous effect of ZEN. A promising way to apply biodegradation in feed is to introduce anaerobic ZEN-degrading microorganisms, which can function during the digestion process in animal intestines. The aim of this study was to isolate anaerobic ZEN-degrading bacteria from anaerobic environments. A strain named F39 was isolated from animal intestinal contents and had a ZEN-degradation rate of 87.35% in 48 h to form trace amount of α- and β-zearalenol. Based on the morphological and physiological properties and phylogenetic analysis of 16S rRNA and *rpoB* gene sequences, F39 was identified as *Clostridium sporogenes*. The optimum temperature for the growth of F39 was 37 °C, the optimum pH was 7.0, and the most suitable carbon source was beef extract, while the optimal conditions for the degradation of ZEN were as follows: 35 °C, pH 7.0, and GAM medium. ZEN was degraded by F39 with a high efficiency in the concentration range of 1–15 mg/L. The bioactive factors responsible for ZEN degradation were mainly distributed intracellularly. F39 can degrade most of the ZEN present, but a small amount is broken down into two secondary metabolites, α- and β-zearalenol, and the toxicity of the degradation products is reduced. With an efficiency of 49%, F39 can more effectively degrade ZEN in wheat-based feedstuffs than in other feedstuff, and the degradation efficiency was pH related. To the best of our knowledge, this is the first report of *Clostridium sporogenes* F39’s ability to maintain the biodegradation potentials.

## 1. Introduction

Zearalenone (ZEN) is a secondary metabolite that was discovered by Stob et al. [1] and is mainly produced by *Fusarium* species, such as *Fusarium graminearum* and *Fusarium cerealis* [2]. *Fusarium* often causes disease in maize, wheat, and other crops and produces mycotoxins, such as zearalenone and deoxynivalenol [3]. ZEN inhibits the proliferation of human T and B lymphocytes and affects the process of immune expression in mononuclear blood cells [4]. ZEN has estrogenic toxicity and reproductive toxicity in rabbits, pigs, and monkeys, affecting their fertility and hormone levels [5]. The effect of ZEN on the activity of specific enzymes in rabbit blood can lead to liver toxicity [6]. The consumption of ZEN-contaminated grains can endanger the health of humans and livestock, as it decreases the expression of CD21^+^ in B cells, increases the proportion of the B1-cell population in pigs, and induces lymphocyte apoptosis, for example [7]. It was also shown that ZEN is carcinogenic and stimulates the growth of MCF-7 human breast cancer cells [8], and the formation of uterine tumors may also be related to ZEN [9].

To reduce the harm of ZEN, various methods of detoxification, such as physical, chemical, and biological methods, are applied. Inorganic montmorillonite can adsorb ZEN, but this process limits the nutritional value of the grain [10]. In comparison, using hydrogen peroxide [11] and ozone [12] to degrade ZEN is more efficient, but the chemical reagents may be toxic and not suitable for practical applications. In terms of microbial detoxification, the cell wall of baker’s yeast was able to adsorb 68% of ZEN [13]. *Lactobacillus plantarum* Lp22, Lp39, and Lp4 were able to degrade ZEN in vitro; however, no degradation product was reported [14]. The *zhd101* hydrolase secreted by *Clonostachys rosea* IFO 7063 was able to open the macrocyclic ester bond in ZEN and form 1-(3,5-dihydroxyphenyl)-10′-hydroxy-1′-undecen-6′-one, which has no estrogenic activity [15]. The yeast *Trichosporon mycotoxinivorans* converts ZEN into (5S)-5-({2,4-dihydroxyl-6-[(1E)-5-hydroxypent-1-en-1-yl]benzoyl}oxy)hexanoic acid (ZOM-1). ZOM-1 showed no estrogenic activity in an in vitro assay. The formation of ZOM-1 was proposed to be a two-step mechanism, involving in the Baeyer–Villiger oxidation on C-6 and the following ring cleavage [16]. The pigs’ intestinal mucosa exhibited activity in reducing ZEN to α-zearalenol (ZEL) and conjugating these metabolites with glucuronic acid [17].

The intestinal microflora is very diverse; for example, one study found that strains isolated from the stomach of cattle can degrade fungal toxins [18], and another study confirmed that microbial strains as well as protozoan populations in the rumen play an effective role in the degradation of ZEN [3]. Since strains were originally isolated from healthy animals and are generally harmless to humans, they have wide application prospects and can be used as food antidotes postprocessing.

At present, researchers have isolated a variety of strains from the natural environment that can efficiently degrade ZEN; these strains include bacteria and fungi and are mostly aerobic. However, because food and feed are digested and decomposed in human and animal intestines, which are normally anaerobic environments, it is difficult for aerobic bacteria to perform a degrading role in practical applications, so it is imperative to isolate anaerobic bacteria that are capable of degrading ZEN. Meanwhile, some intestinal microorganisms such as *Bifidobacterium* and *Lactobacillus* have been shown to have a beneficial effect on human health, and these beneficial microorganisms are termed as probiotics. Their beneficial effects contribute to metabolic products and the interaction with the immune system [19]. Therefore, it would be more efficient to isolate anaerobic degrading bacteria that act both as probiotics and detoxification agents [20]. 

In this study, an anaerobic bacterium F39 that efficiently degraded ZEN was isolated from pig intestinal contents and identified by morphological and phylogenetic analytical tree analysis. The effects of different factors on the growth of F39 and ZEN degradation, including temperature, pH, medium, ZEN concentration, and carbon sources, were also investigated. The ZEN degradation mechanisms were preliminarily explored via LC-TOF-MS/MS analysis. The detoxification effect was evaluated by using cytological experiments and ZEN degradation in feedstuffs to provide new prospects for the degradation of ZEN in feed.

## 2. Materials and Methods

### 2.1. Chemicals and Medium

Authentic ZEN samples were purchased from Merck Co. (Shanghai, China) and dissolved in methanol (10 mg/mL) to form a stock solution in this study. The methanol was of chromatographic grade and purchased from ROE Sciences (Newark, NJ, USA). Authentic α-zearalenol (α-ZEL) and β-zearalenol (β-ZEL) were purchased from Romer Labs (Beijing, China). All other chemicals used were of analytical grade.

The Gifu anaerobic medium (GAM) used contained carcass (10.0 g/L), soybean peptone (3.0 g/L), yeast extract (5.0 g/L), beef extract (2.2 g/L), digested serum powder (13.5 g/L), beef liver extract (1.2 g/L), glucose (3.0 g/L), potassium dihydrogen phosphate (2.5 g/L), sodium chloride (3.0 g/L), soluble starch (5.0 g/L), L-cysteine (0.3 g/L), and sodium thioglycolate (0.3 g/L) with a pH of 7.0. The Luria–Bertani (LB) medium used contained the following components: tryptone (10.0 g/L), yeast extract (5.0 g/L), and sodium chloride (10 g/L) with a pH of 7.0. The minimal medium (MM) used contained 1.6 g/L Na_2_HPO_4_, 1 g/L KH_2_PO_4_, 0.5 g/L NaNO_3_, 0.5 g/L MgSO_4_·7H_2_O, 0.025 g/L CaCl_2_·2H_2_O, and 0.5 g/L (NH_3_)_2_SO_4_ with a pH of 7.0. The anaerobic meat liver broth medium used contained the following components: meat and liver leaching powder (15.0 g/L), peptone (10.0 g/L), sodium chloride (5.0 g/L), and glucose (2.0 g/L). Unless otherwise stated, the medium was supplemented with ZEN just prior to inoculation. The PBS buffer contained 8.0 g/L NaCl, 0.2 g/L KCl, 3.58 g/L Na_2_HPO_4_·12H_2_O, and 0.27 g/L KH_2_PO_4_. All media were autoclaved at 121°C for 20 min before use.

The reagents used for the cytotoxicity assay included the following: the human breast cancer cell line MCF-7 (Xinyu Biological Technology Co., Ltd., Shanghai, China), Dulbecco’s modified Eagle medium (DMEM) (HyClone, Omaha, NE, USA), phenol red–free Dulbecco’s modified Eagle medium (Phenol Red–free DMEM) (Gibco), Solarbio MTT Cell Proliferation and Cytotoxicity Assay Kit, fetal bovine serum (FBS) (Tianhang Biotechnology Co., Ltd., Zhejiang, China), Pen Strep (10,000 U/mL penicillin, 10,000 μg/mL streptomycin, Gibco), and 0.25% trypsin-EDTA (Gibco). An inverted microscope (Nikon ECLIPSE Ti) and a CYTATION3 enzyme-labeled instrument (BioTek Instruments, Inc., Winooski, VT, USA) were used.

### 2.2. Analytical Methods

ZEN in the culture was extracted by partitioning with the same volume of ethyl acetate three times. The organic phases were combined and dried under a nitrogen stream, which was then redissolved in HPLC-grade methanol and passed through a 0.22 μm filter. ZEN was determined by HPLC using the following conditions: Agilent Zorbax SB-C18 column (4.6 × 250 mm, 5 µm); injection volume: 10 μL; mobile phase: methanol:water (80:20 *v*/*v*); flow rate: 0.6 mL/min; UV detection at 236 nm.

The ZEN standard solutions were prepared at concentrations of 1.0, 2.0, 5.0, 10.0, and 20.0 mg/L. Under the above chromatographic conditions, the samples were injected from low to high concentrations, the ZEN peak area A (mAU·S) of each sample was recorded, and the ZEN standard curve was plotted against the concentration (mg/L) of the corresponding standard solution.

### 2.3. Isolation of ZEN-Degrading Strains

Several samples of intestinal contents from livestock were obtained from Jiangsu Academy of Agricultural Sciences, Liuhe Animal Science Base, China. Each sample was suspended in 20 mL of a 0.9% saline solution for 10 min, and the whole operation was carried out in an anaerobic chamber. Then, 0.5 mL of supernatant was placed in 4.5 mL of GAM liquid medium and incubated at 110 rpm in an anaerobic incubator at 37 °C. After 5 d, 0.5 mL was inoculated into 4.5 mL of GAM liquid medium containing 5 mg/L ZEN as the experimental group, and GAM medium without inoculum (containing ZEN) was used as the control. After inoculation and incubation in an anaerobic incubator at 37 °C for 5 d at 110 rpm, the ZEN content was determined by HPLC. The samples that exhibited ZEN degradation effects were selected, and 0.5 mL of each sample was transferred to 4.5 mL of GAM liquid medium that contained 5 mg/L ZEN, and the samples were incubated at 37 °C for 5 d. Then, the ZEN content was measured. After transferring 8 times until the degradation rate of ZEN in each sample was basically stable, the samples were diluted and coated on GAM solid medium and incubated at 37 °C for 5 d. Then, single colonies were picked and inoculated on GAM solid medium and purified three more times. Single colonies were inoculated into GAM liquid medium that contained 5 mg/L ZEN as the experimental group; GAM medium containing the same concentration of ZEN was used as the control group. After incubation at 37 °C for 5 d, the ZEN content was detected by HPLC, and the strains with the ability to degrade ZEN were isolated.

### 2.4. Identification of ZEN-Degrading Strains

ZEN-degrading strains were identified by examining the cell morphology, physiological and biochemical characteristics, and 16S rRNA and rpoB gene sequences. The molecular biology identifications were performed as follows: Bacterial genomic DNA was extracted by using a Qiagen Gentra Puregene Yeast/Bacteria Kit (Germantown, WI, USA). Primers 27F and 1492R (27F: 5′-AGAGTTTGATCCTGGCTCAG-3′; 1492R: 5′-TACCTTGTTACGACTT-3′) were used to amplify the 16S rRNA gene using a routine PCR procedure [21]. The PCR product was purified using the OMEGA Cycle-Pure Kit (100) D6492–01. After purification, the 16S rRNA PCR product was ligated into the pGEM T-Easy vector as recommended by the manufacturer (Promega Corporation, Madison, WI, USA). The sequence was determined by Sangon Biotech (Shanghai, China). The resulting sequences were analyzed through a BLAST search in the NCBI database (http://www.ncbi.nlm.nih.gov/, accessed on 16 August 2021). The recognized standard sequence data with the 16S rRNA source of the strain were obtained from the GenBank database. Phylogenetic trees were constructed by MEGA 7.0 using the neighbor-joining method [22]. The *rpoB* gene sequence was determined by the China General Microbiological Culture Collection Center, and a phylogenetic tree based on the *rpoB* gene sequence was constructed using the same method.

The morphological observations were performed as follows: the screened single bacteria were diluted and coated on GAM solid medium, and the colony morphology was observed after inverted incubation at 37 °C for 5 d.

The cell morphology and physiological and biochemical characteristics were determined by the China General Microbiological Culture Collection Center.

### 2.5. Preparing the Inoculum for Degradation Studies

The inoculum for all experiments was prepared by inoculating the F39 strain in 20 mL of GAM medium and then placing it in an anaerobic sealed chamber on a rotary shaker at 37 °C and 110 rpm for 12 h. The optical density of the bacterial solution at 600 nm (OD_600_) was approximately 1.0. In degradation experiments, cells were inoculated into 50 mL flasks containing 20 mL of GAM and 5 mg/L of ZEN and then incubated at 37 °C and 110 rpm under anaerobic conditions.

### 2.6. Biodegradation of ZEN by F39

#### 2.6.1. Degradation Kinetics of F39

F39 was inoculated into 20 mL GAM liquid medium (containing 5 mg/L ZEN) at 10% inoculum and was incubated at 37 °C for 48 h. The process of ZEN degradation and the growth of strain F39 were detected simultaneously. Samples were taken at set time intervals, and the OD_600_ value of the bacterial solution was measured to quantify the growth and obtain the growth curve of the strain. The concentration of ZEN was determined by HPLC, and the residual ZEN content was plotted as a function of time.

#### 2.6.2. Effect of Culture Conditions on the Growth and Degradation of ZEN by F39

Single-factor optimization was performed for the culture conditions to investigate the effects of different incubation temperatures (25, 30, 35, 37, 40, 45 °C), different initial pH values (3.0, 5.0, 7.0, 9.0, 11.0), different media (MM, LB, anaerobic meat liver broth medium, and GAM), different ZEN concentrations (1, 5, 10, and 15 mg/L), and different carbon sources (glucose, maltose, lactose, fructose, soluble starch, tryptone, yeast extract, and beef extract) on the strain’s growth and ability to degrade ZEN. Samples were taken at regular intervals to quantify the growth by measuring the OD_600_ value of the bacterial solution and to quantify the degradation capacity by measuring the residual ZEN concentration by HPLC. In all experiments, the uninoculated medium was used as the control group.

#### 2.6.3. Localization of the Active Degradation Ingredients in F39

Fifty milliliters of GAM liquid medium was inoculated with 5% F39 culture, and the medium was incubated for 48 h at 37 °C and 110 rpm and then centrifuged at 8000 rpm for 5 min. The supernatant was passed through a 0.22 μm filter membrane and stored at 4 °C. The pellet was washed twice with PBS buffer and resuspended; half of the pellet was ultrasonically disrupted for 30 min and centrifuged at 12,000 rpm for 2 min, and the supernatant was passed through a 0.22 μm filter membrane to afford a cell lysate solution. The other half of the cell pellet was left untouched to produce a cell suspension.

A total of 10 mL of each treated sample was taken, ZEN was added to a level of 5 mg/L, the samples were incubated at 37 °C with 110 rpm shaking for 6 h, samples were obtained, and the ability of each fraction to degrade ZEN was detected by HPLC.

#### 2.6.4. Analysis of the Degradation Products of ZEN by F39

Inoculum from the F39 strain was added to 20 mL of GAM (containing 5 mg/L ZEN) to a final content of 5% and was incubated at 37 °C and 110 rpm for 48 h. One milliliter of GAM culture was taken at 24 and 48 h, extracted twice with an equal volume of ethyl acetate, subjected to nitrogen blowing and drying, redissolved in acetonitrile, and passed through a 0.22 μm filter. GAM without F39 inoculation was used as a blank control. Samples were sent to the Center of Testing and Analysis, Nanjing University, for metabolic profiling by using an AB Sciex 5600^+^ LC-TOF-MS system equipped with an UPLC module and an electrospray ionization (ESI) source. The following gradient elution program was applied: 10% A to 90% A in 9.5 min and retained for 3.5 min, where A was UPLC-grade acetonitrile with 0.1% HCOOH and B was ultrapure water with 0.1% HCOOH. The flow rate was set to 0.2 mL/min. A TOF-MS/MS scan was acquired over a mass range of 100–1200 Da in the positive mode with a collision energy (CE) of 40 eV. 

The obtained raw LC-MS datasets were imported into the MS-DIAL program for spectral deconvolution and peak identification. The alignment of the spectra of three different groups (CK, 24 h, and 48 h after inoculation) was also accomplished in MS-DIAL. The loading plot was used to illustrate the features that were most significantly changed between the CK and degradation groups. 

#### 2.6.5. Degradation of ZEN in Feed by F39

Six feedstuff samples were tested for the biodegradation capacity by F39, including, A: Taohuawu DDGS, Yichun, Jiangsu; B: Jiahui Feed DDGS, Shijiazhuang, Hebei; C: DDGS, Tianguan Biochemical Co., Ltd., Luohe, Henan; D: Jiangyan Wheat Flour, Zhangjiagang, Jiangsu; E: Wheat flour, Zhangjiagang, Jiangsuand; F: Farmhouse corn flour from local residents in Nanjing.

F39 was inoculated in 20 mL GAM liquid medium at 5% inoculum and was incubated for 12 h at 37 °C and 110 rpm with shaking (OD_600_ ≈ 1.0).

Six feeds were used in this study. To determine the ability of F39 to degrade ZEN in feed, 10.0 g of feed was weighed and dispensed into flasks, and 20 mL of sterile water was added and placed in an anaerobic incubator at 37 °C and 130 rpm to degas the feed overnight. Then, 20 mL of bacterial solution was added, and the culture was supplemented with ZEN to a final concentration of 5 mg/L. The culture was then incubated at 37 °C and 130 rpm for 4 d.

ZEN was extracted from the feed and detected by LC–MS as described previously [23].

### 2.7. Cytotoxic Assay of ZEN Degradation Products on MCF-7 Cells

Medium A contained 90% DMEM + 10% FBS + 0.1% Pen Strep. Medium B contained 90% phenol red–free DMEM + 10% CD-FBS + 0.1% Pen Strep. For the preparation of de-estrogenated fetal bovine serum (CD-FBS), dextran (25 mg/100 mL) was added to FBS, and then activated charcoal (250 mg/100 mL) was added after it was completely dissolved in a water bath at 55 °C for 45 min and centrifuged at 3000 rpm for 10 min. The supernatant was obtained and treated 2 consecutive times. The serum was then filtered through 0.22 μm filters and stored at −20 °C. F39 was inoculated into GAM medium with different concentrations of ZEN at 5% inoculum and was incubated at 37 °C and 110 rpm for 48 h. The degradation products were extracted with ethyl acetate and dried with a nitrogen blowing apparatus to obtain ZEN degradation products.

Estrogenic effect assay: Frozen MCF-7 cells were rapidly resuscitated in a 37 °C water bath and cultured in medium A. When the cells had grown to 80–90% confluence in the culture flask, they were digested with 0.25% trypsin-EDTA and transferred to medium B. After two days of culture, they were digested with 0.25% trypsin-EDTA and then made into single-cell suspensions of 10^5^ cells/mL using medium B. The cells were inoculated into 96-well plates at a concentration of 2 × 10^4^ cells/well in a volume of 100 μL. To adhere the cells to the wall, 12 h of incubation was performed, then the intervention factors were added to each well of the experimental group at concentrations of 10^−9^, 10^−8^, 10^−7^, 10^−6^, and 10^−5^ mol/L for ZEN or the degradation products of ZEN, and estradiol (E2) was set at 10^−9^ mol/L as a positive control. The negative control group was incubated in medium B without ZEN and the degradation products of ZEN. Six replicates were set for each group. The staining time was 24 h. After 24 h, the cell morphology was observed with an inverted microscope and was photographed. The staining solution was discarded, and the cells were stained with the MTT kit (MTT Cell Proliferation and Cytotoxicity Assay Kit, Solarbio, Beijing, China). The absorbance value (A) of each well was measured at 490 nm, which was selected by an enzyme-labeled instrument, and the relative proliferation rate (PR) of each group was calculated as follows:

PR = (A of experimental group/A of control group) × 100%.Cell culture was performed in an incubator at 37 °C with 5% CO_2_.

Toxicity testing of degradation products: MCF-7 cells were incubated as described above. Authentic ZEN, authentic E2 and the ethyl-acetate extracted degradation products were all dissolved in DMSO to a concentration of 1000 mg/L, respectively, and then added to the cell culture to a certain concentration. Concentration levels of 1, 5, 10, 15, 20, and 25 mg/L were used for both ZEN and the degradation products of ZEN, and the negative control was culture B without ZEN or ZEN degradation products.

### 2.8. Statistical Analysis

All results are expressed as the means of at least three replicates with their standard deviations. One-way ANOVA was performed using GraphPad Prism 8.4.2 [24], and *p* < 0.05 indicated significant differences.

## 3. Results and Discussion

### 3.1. Isolation and Identification of ZEN-Degrading Strains

Four strains that were capable of degrading ZEN, namely, F4, F8, F15, and F39, were screened from colonies on GAM plates. Strain F39 had the best ZEN-degrading activity, as it degraded ZEN from 5.0 to 0.63 mg/L within 48 h with a degradation rate of 87.35%. The degradation rates of the remaining strains were lower than that of F39. Therefore, F39 was selected for further study.

On GAM plates, F39 can form visible colonies, which are somewhat round, translucent, and white to yellowish; have uneven margins; spread outward; and often spread into moss. After 72 h of incubation at 37 °C, the colony diameter was 1.0–3.0 cm (Figure 1A). F39 is a rod-shaped Gram-positive bacterium with a diameter of approximately 0.40–0.60 × 3.50–6.50 μm. In addition, F39 is capable of producing spores, has ovoid-shaped budding spores, is wider than the propagule, is located at the secondary extremities, and has cells that are shaped like tennis racket (Figure 1B).

The 16S rRNA and *rpoB* gene sequences of strain F39 were determined. The 16S rRNA gene sequences had the highest identity (100%) with those of the *Clostridium botulinum* archived in GenBank. A phylogenetic tree based on the 16S rRNA gene sequences showed that F39 was very closely related to *Clostridium botulinum* CP027776.1 and *Clostridium sporogenes* LC145546.1 (Figure 1C). Further phylogenetic analysis of the *rpoB* gene showed that F39 was closely related to *Clostridium botulinum* CP027780.1 and *Clostridium sporogenes* CP013242.1 (Figure 1D).

Because we were unable to distinguish the strain based solely on 16S rRNA and *rpoB* gene sequences, the physiological and biochemical properties of F39 were further validated (Table 1). According to the phylogenetical analysis, F39 was also close to *C. botulinum*, which is notorious for causing botulism. However, F39 showed different physiochemical characteristics from those of *C. botulinum*. For instance, F39 was able to utilize D-glucose and maltose, while *C. botulinum* was unable to ferment on such carbohydrates. Therefore, F39 was identified as *Clostridium sporogenes* based on the analysis of its cell morphology, physiological and biochemical characteristics, and 16S rRNA and *rpoB* gene sequences, with reference to *Berger’s Manual of Systematic Bacteriology* and the *International Journal of Systematic and Evolutionary Microbiology*.

*Clostridium sporogenes* is a well-targeted, safe, and self-limiting target vector that is used for solid tumor therapy. Budd et al. found that using *Clostridium sporogenes* as a delivery vehicle for cancer therapy offers the possibility of treating cancer, and this treatment kill cancer cells by utilizing the ability of spores to grow in tumors [25]. Ionata et al. isolated *Clostridium sporogenes* from solfataric muds and found that *Clostridium sporogenes* could grow on chicken feathers and exhibited keratinolytic activity [26]. The keratinase secreted by *Clostridium sporogenes* has the unique ability to degrade chicken feathers. *Clostridium sporogenes* is also often used as a model strain for sterilization experiments due to its ability to produce spores [27]. 

Recently, the use of microorganisms to adsorb or degrade ZEN has become an ideal and efficient means to eliminate ZEN, and several microorganisms have been isolated with the purpose of eliminating ZEN. Wang et al. isolated *Bacillus pumilus* ES-21 from soil, which showed a 95.7% degradation rate of ZEN [28]. Yang et al. identified a novel biotransformation mode of ZEN into ZEN-14-phosphate by the strain *Bacillus subtilis* Y816 [29]. Brodehl et al. found that several *Aspergillus oryzae* strains and *Rhizopus* species were able to convert ZEN into ZEN-14-sulfate as well as ZEN-O-14-, ZEN-O-16-glucoside, and α-zearalenol (α-ZEL) [30]. Additionally, a novel fungal metabolite, α-ZEL-sulfate, was detected. Tan et al. screened two *Pseudomonas* strains from soil, and these strains also had ZEN-degrading effects [31]. Recently, Budd et al. discovered the use of *Clostridium sporogenes* as a delivery vehicle for cancer therapy, and they used the ability of the spores to grow in tumors to kill cancer cells, offering the possibility of treating cancer [25]. However, no ZEN-degrading *Clostridium sporogenes* has been reported, and the discovery of this strain opens a new pathway for the degradation of ZEN. Furthermore, F39 has some research value in the degradation of ZEN.

*Clostridium* spp. have a huge potential as probiotics according to previous studies, indicating that F39, explored in this study, was able to maintain the intestinal health for stock animals as well as to eliminate the toxicity caused by ZEN [32]. 

### 3.2. Effect of Culture Conditions on the Growth of F39

The effects of temperature, pH, and different carbon sources on the growth of F39 are shown in Figure 2. As shown in Figure 2A, F39 was able to proliferate over a wide range of temperatures. The fastest growth rate and the maximum OD_600_ value (1.96) were observed within 48 h at 37 °C. The growth rates were slower at 35, 40, and 45 °C than at 37 °C, and the final bacterial density was also less than that at 37 °C. At 25 and 30 °C, the growth of F39 was slower. The optimal growth temperature for F39 was 37 °C.

Strain F39 grew well from pH 7.0 to pH 11.0 (Figure 2B). Moreover, pH 7.0 was the optimum pH for the growth of F39, with a maximum OD_600_ of 1.9033. F39 grew better under alkaline conditions and was inhibited under acidic conditions at pH 3.0.

Different carbon sources were added to the MM medium. The growth of F39 differed with different carbon sources (Figure 2C). F39 grows slowly and has a low biomass under glucose, fructose, maltose, and starch conditions. The growth of F39 in lactose, beef extract, peptone, and yeast extract was better than that in the above four carbon sources, but the final OD_600_ did not exceed 1.0 even after 48 h. F39 showed maximum growth when beef extract was used as a carbon source.

### 3.3. Effect of Culture Conditions on the Degradation Efficiency of F39

The degradation kinetics of ZEN and the growth of F39 were monitored within 48 h of inoculating F39 into the ZEN-containing GAM (Figure 3A). The majority of ZEN was degraded, and the residual ZEN concentration decreased from 4.27 to 0.46 mg/L (the degradation rate was 89.31%). In the process of ZEN degradation, F39 was in the logarithmic growth phase from 10 to 20 h, with the cell concentration rising from 1.8 × 10^8^ to 4.74 × 10^8^ CFU/mL; the growth decreased slowly after 30 h, and the cells entered the decline phase (from 4.82 × 10^8^ CFU/mL at 30 h to 4.42 × 10^8^ CFU/mL at 48 h).

The effects of temperature, pH, medium, and substrate concentration on the degradation of ZEN by F39 are shown in Figure 3. ZEN can be degraded from 25 to 45 °C (Figure 3B). At temperatures below 35 °C, the degradation rate increases with increasing temperature. In the first 24 h, the degradation rate of ZEN by F39 at 40 °C was slightly higher than that at 45 °C, after which the degradation rate was accelerated at 45 °C, and finally, the degradation rate at 45 °C was higher than that at 40 °C. The optimal temperature for ZEN degradation was 35 °C, at which F39 had the highest degradation rate (85.79%).

F39 showed good efficiency of ZEN degradation in the pH range of 7.0–11.0 (Figure 3C). The F39 strain showed almost no ZEN degradation at pH 3.0; a slightly lower degradation rate of 64.79% was observed at pH 5.0 after 48 h. F39 showed the highest growth and degradation rate of 82.33% at pH 7.0; there was no significant difference between the degradation capacities at pH 9.0 and 11.0.

The effect of different media on the ZEN degradation rates by F39 was significant (Figure 3D). ZEN remained almost intact in MM, indicating that either F39 could not use ZEN as its sole carbon source or the ZEN content dissolved in solution was not sufficient as a carbon source. In anaerobic meat liver broth, ZEN degraded slowly in the first 24 h, and the final degradation rate was lower than that of GAM. F39 multiplied rapidly in GAM, and the degradation rate was high, reaching 85.73%. The GAM medium was more suitable for both the growth of F39 and the degradation of ZEN.

The degradation of ZEN by F39 was also related to the substrate concentration (Figure 3E). In the initial ZEN concentration range of 1.0–10.0 mg/L, the final ZEN degradation rates did not greatly vary and were all approximately 90%. At a ZEN concentration of 15 mg/L, the degradation rate was significantly lower than that of the other four groups at the same time, but the absolute reduction in ZEN still showed an increasing trend.

### 3.4. Mechanism of ZEN Degradation by Strain F39

To locate the active degradation components, we analyzed the ZEN residues in different fractions from the culture (Figure 4A). The cell lysate solution showed the best degradation capacity, reaching a degrading rate of 77.30% when the concentration of ZEN substrate was 5.0 mg/L. Meanwhile, the degradation catalyzed by the supernatant of F39 culture was not as efficient as that of the cell lysate solution, presenting a 17.73% degradation rate. In addition, ZEN in the bacterial suspension was reduced by 14.20%, which was much lower than the reduction in ZEN in the cell lysate solution. It can be determined that cell adsorption is not the main reason for the reduction in ZEN by F39 and that the active degradation components are mainly distributed intracellularly.

To study the degradation mechanism of ZEN by F39, the degradation products were analyzed by LC-TOF-MS. It was found that after degradation by F39, the metabolic profiles were significantly different between the degradation and control groups. After degradation, several ZEN-related new MS features emerged. However, the low abundance of such degradation products hindered our efforts to purify and identify them by NMR. The only two degradation products were α-ZEL and β-ZEL by interpreting their MS/MS patterns and comparison with authentic samples. The other features were unable to be elucidated because they lack MS/MS spectra due to the low mass intensity. The ratio of α-ZEL to β-ZEL was determined to be approximately 1:2 by calculating the peak area and comparing it with authentic compounds (Appendix A). Thus, one possible degradation route was that F39 reduced ZEN into α-ZEL and β-ZEL (Figure 4B). However, the amounts of both metabolites were much lower (72 μg/L for α-ZEL and 136 μg/L for β-ZEL) than the amount of ZEN added (5 mg/L); it is likely that only a small part of ZEN had decomposed into α-ZEL and β-ZEL, and most of these compounds were completely degraded, indicating that F39 was efficient in eliminating ZEN content. 

A total of 15 ZEN derivatives have been identified, and the derivatives include α-ZEL, β-ZEL, zearalanone (ZAN), α-ZAL, β-ZAL, ZEN-14G, ZEN-16G, ZEN-GlcA, ZEN-sulfate, α-ZEL-14G, β-ZEL-14G, α-ZAL-14G, β-ZAL-14G, ZAN-14G, and ZAN-16G. However, only two degradation products were identified by LC-TOF-MS and determined to be α-ZEL and β-ZEL in this study. ZEN is mainly metabolized to α-ZEL in humans, pigs, and mice and to β-ZEL in poultry and ruminants [33]. The estrogenic potency of α-ZEL is more than 500-fold that of ZEN, whereas β-ZEL has an estrogenic potency 16-fold lower than that of ZEN [34,35]. Therefore, we further explored the toxic effects of the degradation products of ZEN.

### 3.5. Estrogenic and Cytotoxic Effects of the ZEN Degradation Products on MCF-7 Cells

Because ZEN has estrogenic effects [36,37], the human breast cancer cell line MCF-7 with estrogen receptor (ER) was selected for cellular experiments. ZEN and its degradation products had the following estrogenic effect on MCF-7 cells (Figure 5A): E2 promoted cell growth, and the relative proliferation rate of MCF-7 cells was 118.43% at an E2 concentration of 10^−9^ mol/L. The growth of MCF-7 cells was promoted by ZEN at 10^−9^–10^−7^ mol/L, and cell proliferation was inhibited by ZEN at 10^−6^–10^−5^ mol/L. The ZEN degradation products promoted the growth of MCF-7 cells in the low concentration range (10^−9^–10^−5^ mol/L), and the relative proliferation rate was higher than that in the ZEN group, with a maximum value of 108.09% at 10^−6^ mol/L. However, considering that the majority of ZEN was completely eliminated, F39 still significantly reduced the harmful effects of ZEN on human health.

The toxic effects of ZEN and its degradation products on MCF-7 cells were further examined. Figure 5B(I) shows the untreated MCF-7 cells, which possess a high number of cells that are polygonal and hyaline and adhere to the wall. Figure 5B(II) shows the MCF-7 cells treated with 25 mg/L ZEN. At this concentration, large numbers of these cells were killed after treatment; the number of cells decreased significantly, the cells fell off of the wall, and the cells became round. Figure 5B(III) shows the MCF-7 cells treated with 25 mg/L ZEN degradation products. Some cells were killed, and the number of cells observed under the microscope decreased, but the density was significantly higher than that in Figure 5B(II). It can also be seen from Figure 5C that in the high-concentration range (1–25 mg/L, or 3.13 × 10^−6^ to 78.37 × 10^−6^ mol/L), different concentrations of ZEN and the degradation products of ZEN showed different toxicities to MCF-7 cells; higher concentrations were more toxic to MCF-7 cells and killed more cells. However, with the ZEN treatment, more cells died at the same concentration, meaning that the ZEN degradation products were less toxic at the same concentration. It is assumed that F39 can degrade ZEN efficiently, and because the majority of ZEN was not converted into ZEL, the toxicity is lower for the degraded product than the original toxin, which provides a basis for the safety of its use.

### 3.6. Degradation of ZEN in Feedstuffs by F39

The effect of ZEN degradation by F39 differed among the different feeds (Table 2). F39 had a better ZEN-degradation effect in wheat feed, with feed D reaching 48.57% and feed E reaching 43.59%. The ZEN degradation rate in DDGS feed was lower than that of wheat, and the degradation rates of ZEN in feeds A, B, and C were 25.79%, 18.81%, and 25.32%, respectively. The ZEN-degradation rate in corn flour was 38.04%. To clarify the reason for the low ZEN-degrading rate in DDGS by F39, the pH value of different feeds was measured (Table 2), and it was found that the pH of DDGS was approximately 4.0, which was lower than the optimum growth pH of F39. The pH of feed sample A was adjusted to 7.0, and then the ZEN degradation rate was measured by F39. The results show that the degradation rate was 43.12%, which is significantly higher than the degradation effect before pH adjustment (Table 2). The acidity of the DDGS feed results in a lower degradation rate. The ZEN-degrading enzymes in F39 performed better under alkaline conditions, as described before; however, corn and its byproduct feeds are often acidic, which affects the degradation ability of the strains. Therefore, the pH value of degrading bacteria can be adapted and optimized in practical applications to broaden the usage of degrading bacteria.

## 4. Conclusions

In this study, we isolated *Clostridium sporogenes* F39, an anaerobic Gram-positive bacterium from the content of pig intestines that can efficiently degrade ZEN, with a maximum bacterial growth of OD_600_ = 2.8 at 37 °C under anaerobic conditions and a degradation rate of 87.35% for ZEN. We obtained the following results from the exploratory experiments: The optimal temperature for the growth of F39 was 37 °C, the optimal pH was 7.0, and the most suitable carbon source for its growth was beef extract. For the degradation of ZEN by F39, the optimal temperature was 35 °C, the optimal pH was 7.0, and the best medium was GAM. F39 could effectively degrade ZEN in the concentration range of 1 to 15 mg/L. The active degradation ingredient in F39 was mainly distributed intracellularly. Most of the ZEN present was completely degraded during the degradation process, and a small proportion was broken down into two ZEN derivatives, α-zearalenol (α-ZEL) and β-zearalenol (β-ZEL); compared to ZEN, the degradation products were much less toxic to MCF-7 cells. F39 degrades ZEN better in wheat feedstuffs and less effectively in maize and DDGS feedstuffs, with a degradation rate of nearly 50% in wheat. The degradation efficiency of F39 in feedstuffs is pH-related and can be improved by adjusting the pH of the feed to a pH suitable for F39 growth.

## Figures and Tables

**Figure 1 foods-11-01194-f001:**
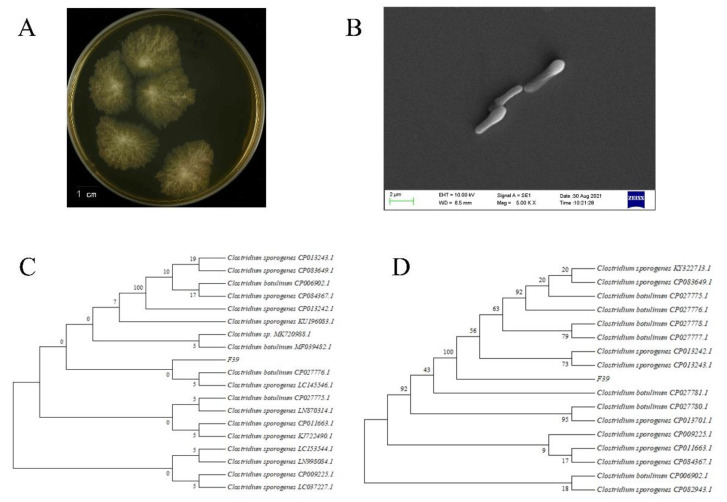
Characterization of F39. (**A**) Colony morphology of F39 on GAM. (**B**) Scanning electron micrograph of F39. (**C**) Phylogenetic tree of F39 based on 16S rRNA gene sequences. (**D**) Phylogenetic tree of F39 based on *rpoB* gene sequences.

**Figure 2 foods-11-01194-f002:**
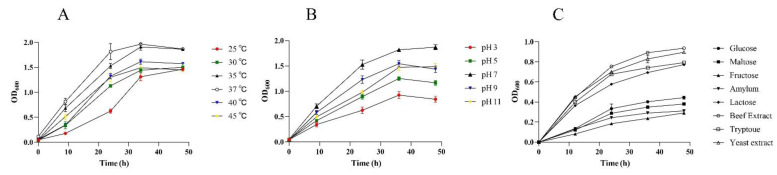
Effect of temperature (**A**), pH (**B**), and different carbon sources (**C**) on F39 growth. OD_600_ values were used to present the growth rate of F39.

**Figure 3 foods-11-01194-f003:**
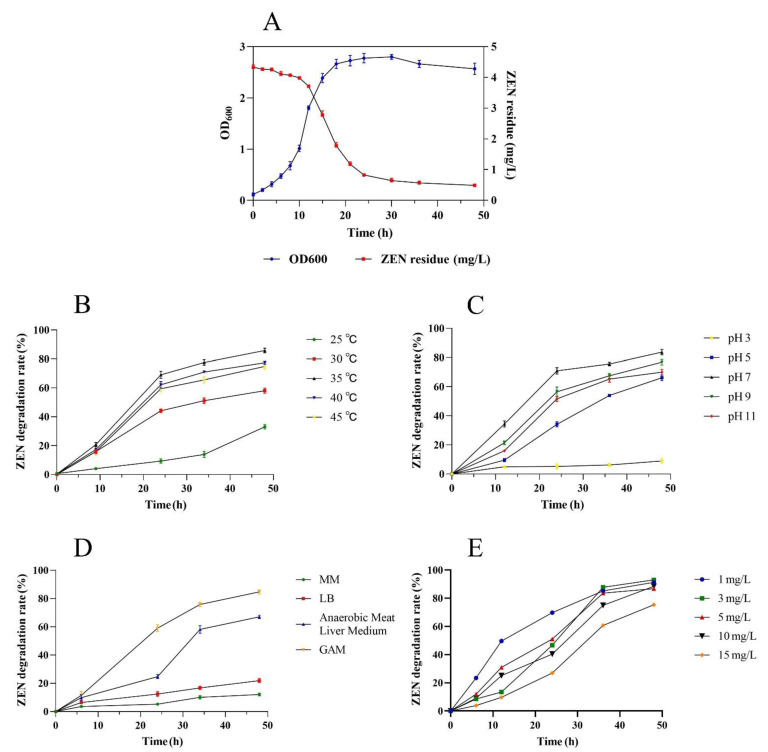
The growth rate of F39 and degradation of ZEN during the culture (**A**). F39 cells were inoculated at the 10% level into 20 mL of GAM containing 5 mg/L ZEN and were then cultured at 37 °C and 110 rpm in an anaerobic environment. The effect of temperature (**B**), pH (**C**), cultural media (**D**), and substrate concentration (**E**) on the degradation of ZEN by F39.

**Figure 4 foods-11-01194-f004:**
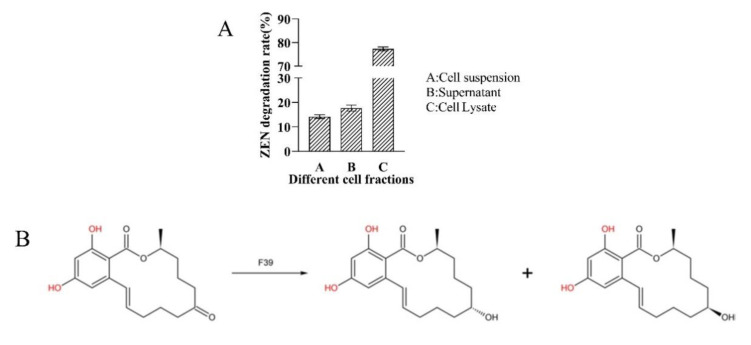
ZEN degradation by different cell components of F39 (**A**) and one possible route of ZEN degradation (**B**).

**Figure 5 foods-11-01194-f005:**
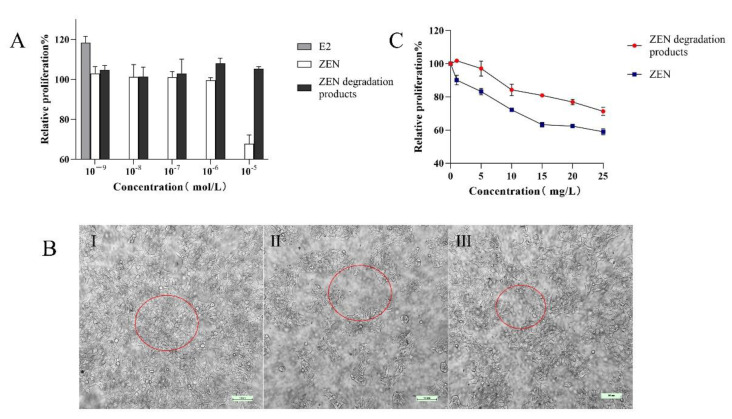
Estrogenic effects of E2, ZEN, and the degradation products of ZEN on the proliferation (**A**) and cell morphology (**B**) of MCF-7 cells (I: untreated MCF-7 cells; II: MCF-7 cells treated with 25 mg/L of ZEN; III: MCF-7 cells treated with 25 mg/L of ZEN degradation products). The toxic effect of substrate concentration of ZEN and the degradation products on the proliferation of MCF-7 cells (**C**).

**Table 1 foods-11-01194-t001:** Cellular morphological and physiochemical characteristics of F39.

Experimental Projects	Results	Experimental Projects	Results	Experimental Projects	Results
Cell morphology	Rods	Gram stain	Positive	Oxidase	−
Catalase from *Micrococcus lysodeikticus*	−	Nitrate reduction	−	Indole production	−
Acid production from fermented glucose	+	Arginine dihydrolase	+	Urease	+
Heptachloride hydrolysis	+	Gelatin hydrolysis	+	β-Galactose adenosine	−
Acid production from carbohydrates (API 50CH)					
Glycerin	+	Inositol	−	Inulin	−
Erythritol	−	Mannitol	−	Melezitose	−
d-Arabinose	−	Sorbitol	−	Raffinose	−
l-Arabinose	−	a-Methyl-d-mannoside	−	Starch	+
d-Ribose	−	a-Methyl-d-glucoside	+	Glycogen	+
d-Xylose	−	N-Acetyl-glucosamine	+	Xylitol	−
l-Xylose	−	Amygdalin	−	Gentiobiose	−
Adonol	−	Arbutin	+	d-Sondiose	+
β-Methyl-d-xyloside	−	Esculin hydrate	+	d-Lysose	−
d-Galactose	−	Salicin	+	d-Tagatose	+
d-Glucose	+	Fibrous disaccharides	+	d-Fucose	−
d-Fructose	+	Maltose	+	l-Fucose	−
d-Mannose	−	Lactose	−	d-Arabinitol	−
l-Sorbose	−	Melibiose	−	l-Arabinitol	−
l-Rhamnose	−	Sucrose	−	Gluconate	−
Dulcitol	−	d-Trehalose anhydrous	+	2-Keto-gluconate	−

**Table 2 foods-11-01194-t002:** Degradation of ZEN in different feedstuffs by F39 and the pH values of each feedstuff.

Feed Type	A	A (pH Modified)	B	C	D	E	F
ZEN degradation rate (%)	25.79 ± 1.76	42.39 ± 2.91	18.81 ± 0.46	25.32 ± 0.55	48.65 ± 2.58	43.59 ± 3.39	38.04 ± 1.42
pH	4.18 ± 0.02	7.00 ± 0.02	3.79 ± 0.03	4.05 ± 0.03	4.99 ± 0.03	5.11 ± 0.06	4.75 ± 0.22

## Data Availability

The data presented in this study are available on request from the corresponding author.

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
