# Peer review of "Isolation, Characterization, and Application of Clostridium sporogenes F39 to Degrade Zearalenone under Anaerobic Conditions"

_foods, 2022, doi:10.3390/foods11091194_

Round 1
Reviewer 1 Report
In the present manuscript, Zhai et al., highlighted the isolation of a new strain, Clostridium sporogenes F39, capable to degrade zearalenone (ZEN) under anaerobic conditions. The strain was isolated from animal intestinal contents, and identified based both on the morphological and physiological properties and on phylogenetic analysis after 16S rRNA and rpoB genes sequence. Furthermore, the authors optimized the conditions for zearalenone degradation in terms of temperatures, pH and carbon sources. The research has been completed by the mechanism of ZEN degradation and the cytotoxicity test using MCF-7 cell line.
The article is well written and structured. The results obtained are very well presented and interpreted. I only have minor comments:
- Please, check the sentence from the line 160 and rephrase because the bacterial DNA is not extracted using primers (“Bacterial genomic DNA was extracted by Primers 27F, and 1492R (27F: 5'-AGAGTTTGATCCTGGCTCAG-3'…. “).
- Please, check the sentence from the lines 192-193 and rephrase because the standard curve was plotted for the ZEN and not for the strain degradation. (“….. and the degradation curve of the strain was plotted”).
- Please, specify which are exactly all the six feeds tested (line 232, materials and methods) in order to be in concordance with table 2 from the results.
- Please, write the manufacturer for the MTT kit (line 264).
- It is important to mention the F39 cell concentration for the optimized conditions of the ZEN degradation. I read that (line 343) the bacterium was under stationary/decline faze, but is important to have a cell concentration expressed in CFU/mL.
- In the conclusion sub-chapter, the first paragraph (from line 459 to 472) should be inserted in the results and discussion sub-chapter because there are presented results from the literature.
- I also recommend to read and to take into consideration for the discussion chapter, the following article: New Biotransformation Mode of Zearalenone Identified in Bacillus subtilis Y816 Revealing a Novel ZEN Conjugate. Shi Bin Yang, et al., 2021.
Author Response
Dear Editor,
Many thanks for your mail on Mar 31th concerning the revision of our manuscript entitled ' Isolation, characterization, and application of Clostridium sporogenes F39 to degrade zearalenone under anaerobic conditions ' (Manuscript ID foods-1654857). Our sincere gratitude must be extended to the reviewers for their positive comments and suggestions, which are very valuable for our improvement of the manuscript. We have carefully gone over all these comments and suggestions and attempted to fully take them into this revision. Now I am pleased to send you the revised version of the manuscript in which the modifications were highlighted in yellow.
Sincerely,
Gang Wang
Response to Reviewer 1 Comments
Point 1: Please, check the sentence from the line 160 and rephrase because the bacterial DNA is not extracted using primers (“Bacterial genomic DNA was extracted by Primers 27F, and 1492R (27F: 5'-AGAGTTTGATCCTGGCTCAG-3'…. “).
Response 1: Revised as suggested.
Point 2: Please, check the sentence from the lines 192-193 and rephrase because the standard curve was plotted for the ZEN and not for the strain degradation. (“….. and the degradation curve of the strain was plotted”).
Response 2: The sentence was revised into “the residue ZEN content was plotted as a function of time.” to make it more clear.
Point 3: Please, specify which are exactly all the six feeds tested (line 232, materials and methods) in order to be in concordance with table 2 from the results.
Response 3: Done as suggested.
Point 4: Please, write the manufacturer for the MTT kit (line 264).
Response 4: Done as suggested.
Point 5: It is important to mention the F39 cell concentration for the optimized conditions of the ZEN degradation. I read that (line 343) the bacterium was under stationary/decline faze, but is important to have a cell concentration expressed in CFU/mL.
Response 5: The cell density of F39 was quantified in CFU/ml during the cultivation, and the result was added to the revised manuscript.
Point 6: In the conclusion sub-chapter, the first paragraph (from line 459 to 472) should be inserted in the results and discussion sub-chapter because there are presented results from the literature.
Response 6: Revised as suggested.
Point 7: I also recommend to read and to take into consideration for the discussion chapter, the following article: New Biotransformation Mode of Zearalenone Identified in Bacillus subtilis Y816 Revealing a Novel ZEN Conjugate. Shi Bin Yang, et al., 2021.
Response 7: The mentioned literature and some related publications were cited in the revised version.

Reviewer 2 Report
This study aims to characterize an anaerobic bacterial strain capable of degrading zearalenone. The main interest of the work is the anaerobic character of the strain used. The main weakness of the manuscript is the lack of structural characterization of the degradation products formed. The bioassay used to characterize the estrogenic effects of the degradation products formed partly circumvents the problem. Below are comments and suggestions for improvement of the manuscript
Abstract : « A strain named F39 was isolated from animal intestinal contents and had a ZEN-degradation rate of 87.35% in 48 h.” clarify here the degradation formed products.
Introduction, paragraph on aerobic strains “At present, researchers … of degrading ZEN” : This is a key aspect of the interest of anaerobic germs as pre-biotics, so please elaborate a bit on this point and add references.
Add few information on the degradation products that are reported in the literature depending on the bacterial strain used.
Material and methods: clarify the provider of alpha-zearalenol and beta-zearalenol, add few information on the method of analysis, notably what was done to characterize the biodegradation product.
Results, 3.4 Mechanism of ZEN degradation by strain F39: please provide more details on the results observed after LC-TOF-MS: which metabolites found? Quantities? Figure S1 provides only results about α-ZEL and β-ZEL, what about the other metabolites?
You said that “After degradation, several ZEN-related metabolites emerged, and the most abundant were α-zearalenol (α-ZEL) and β-zearalenol (β-ZEL).” it would be pleasant to have some quantitative results, at least on these 2 metabolites.
You said “A total of 15 ZEN derivatives have been identified,” which among them were found as degradation products in this study?
3.6 Degradation of ZEN in feedstuffs by F39, lane 6, check for typing error
Conclusion is too long, focus on key results and perspectives
Author Response
Dear Editor,
Many thanks for your mail on Mar 31th concerning the revision of our manuscript entitled ' Isolation, characterization, and application of Clostridium sporogenes F39 to degrade zearalenone under anaerobic conditions ' (Manuscript ID foods-1654857). Our sincere gratitude must be extended to the reviewers for their positive comments and suggestions, which are very valuable for our improvement of the manuscript. We have carefully gone over all these comments and suggestions and attempted to fully take them into this revision. Now I am pleased to send you the revised version of the manuscript in which the modifications were highlighted in yellow.
Sincerely,
Gang Wang
Response to Reviewer 2 Comments
Point 1: Abstract : « A strain named F39 was isolated from animal intestinal contents and had a ZEN-degradation rate of 87.35% in 48 h.” clarify here the degradation formed products.
Response 1: Revised as suggested.
Point 2: Introduction, paragraph on anaerobic strains “At present, researchers … of degrading ZEN” : This is a key aspect of the interest of anaerobic germs as pro-biotics, so please elaborate a bit on this point and add references.
Response 2: Intestinal microorganisms like Bifidobacterium and Lactobaciillus have been shown to have a beneficial effect on human health and are called probiotics. Clostridium spp. have a huge potential as probiotics according to previous studies, indicating that F39 in this study was able to maintain the intestinal health for stock animals as well as to eliminate the toxicity caused by ZEN. Relating information was issued in the revised manuscript.
Point 3: Add few information on the degradation products that are reported in the literature depending on the bacterial strain used.
Response 3: Information regarding the biodegradation products was cited in the Introduction section.
Point 4: Material and methods: clarify the provider of alpha-zearalenol and beta-zearalenol, add few information on the method of analysis, notably what was done to characterize the biodegradation product.
Response 4: Authentic α-zearalenol (α-ZEL) and β-zearalenol (β-ZEL) were purchased from Romer Labs (Beijing, China). The detailed LC-TOF-MS method for the analysis of degradation products was added to the revised manuscript.
Point 5: Results, 3.4 Mechanism of ZEN degradation by strain F39: please provide more details on the results observed after LC-TOF-MS: which metabolites found? Quantities? Figure S1 provides only results about α-ZEL and β-ZEL, what about the other metabolites?
Response 5: After biodegradation of ZEN by F39, several features emerged according to LC-TOF-MS analysis. However, the low abundance of such degradation products hindered our efforts to the purification and identification of them by NMR. The only two degradation products were identified to be α-ZEL and β-ZEL by intepreting their MS/MS patterns and comparison with authentic samples. The other features were unable to be elucidated because they lack MS/MS spectra due to the low mass intensity.
Point 6: You said that “After degradation, several ZEN-related metabolites emerged, and the most abundant were α-zearalenol (α-ZEL) and β-zearalenol (β-ZEL).” it would be pleasant to have some quantitative results, at least on these 2 metabolites.
Response 6: The amounts of α-ZEL and β-ZEL were determined to be , respectively. Although α-ZEL and β-ZEL were the most abundant product that emerged after biodegradation by F39, their amount was not comparable to the amount of ZEN substrate, indicating that F39 was efficient in eliminating ZEN content.
Point 7: You said “A total of 15 ZEN derivatives have been identified,” which among them were found as degradation products in this study?
Response 7: According to Wu’s review, ZEN derivatives transformed by poultry intestinal microorgansim were α-zearalenol, β-zearalenol, zearalanone, α-zeralanol, β-zeralanol, ZEN-14G, ZEN-16G, ZEN-GlcA, ZEN-Sulphate, α-zearalenol-14G, β-zearalenol-14G, α-zearalanol-14G, β-zearalanol-14G, zearalanone-14G and zearalanone-16G (Wu, K.; Ren, C.; Gong, Y.; Gao, X.; Rajput, S.A.; Qi, D.; Wang, S. The Insensitive Mechanism of Poultry to Zearalenone: A Review. Anim. Nutr. 2021, 7, 587–594.). However, the only two degradation products detected by LC-TOF-MS were identified to be α-ZEL and β-ZEL in this study.
Point 8: 3.6 Degradation of ZEN in feedstuffs by F39, lane 6, check for typing error
Response 8: Done as suggested.
Point 9: Conclusion is too long, focus on key results and perspectives.
Response 9: Conculsion was revised to make more clarity.
